# Free Fatty Acids Are Associated with the Cognitive Functions in Stroke Survivors

**DOI:** 10.3390/ijerph18126500

**Published:** 2021-06-16

**Authors:** Dariusz Kotlęga, Barbara Peda, Joanna Palma, Agnieszka Zembroń-Łacny, Monika Gołąb-Janowska, Marta Masztalewicz, Przemysław Nowacki, Małgorzata Szczuko

**Affiliations:** 1Department of Neurology, District Hospital, 67-200 Glogow, Poland; barbarapeda@op.pl; 2Department of Human Nutrition and Metabolomics, Pomeranian Medical University, 71-460 Szczecin, Poland; joanna.palma@pum.edu.pl (J.P.); malgorzata.szczuko@pum.edu.pl (M.S.); 3Department of Applied and Clinical Physiology, Collegium Medicum, University of Zielona Gora, 65-001 Zielona Góra, Poland; a.zembron-lacny@cm.uz.zgora.pl; 4Department of Neurology, Pomeranian Medical University, 71-252 Szczecin, Poland; monikagj@op.pl (M.G.-J.); nowiczontko@poczta.onet.pl (M.M.); przemyslaw.nowacki@pum.edu.pl (P.N.)

**Keywords:** ischemic stroke, dementia, risk factors, saturated fatty acids, polyunsaturated fatty acids, inflammation, lipids, cerebrovascular diseases, omega-3, cognitive decline, neuropsychological functions

## Abstract

Ischemic stroke is a leading cause of motor impairment and psychosocial disability. Although free fatty acids (FFA) have been proven to affect the risk of stroke and potentially dementia, the evidence of their impact on cognitive functions in stroke patients is lacking. We aimed to establish such potential relationships. Seventy-two ischemic stroke patients were prospectively analysed. Their cognitive functions were assessed seven days post-stroke and six months later as follow-up (*n* = 41). Seven days post-stroke analysis of serum FFAs levels showed direct correlations between Cognitive Verbal Learning Test (CVLT) and the following FFAs: C20:4n6 arachidonic acid and C20:5n3 eicosapentaenoic acid, while negative correlations were observed for C18:3n3 linolenic acid (ALA), C18:4 n3 stearidonic acid and C23:0 tricosanoic acid. Follow-up examination with CVLT revealed positive correlations with C15:0 pentadecanoid acid, C18:3n6 gamma linoleic acid, SDA, C23:0 tricosanoic acid and negative correlations with C14:0 myristic acid and C14:1 myristolenic acids. Several tests (Trail Making Test, Stroop Dots Trail, Digit Span Test and Verbal Fluency Test) were directly correlated mainly with C14:0 myristic acid and C14:1 myristolenic acid, while corresponding negatively with C18:1 vaccinic acid, C20:3n3 cis-11-eicosatrienoic acid, C22:1/C20:1 cis11- eicosanic acid and C20:2 cis-11-eicodienoic acid. No correlations between Montreal Cognitive Assessment (MOCA) test performed on seventh day, and FFAs levels were found. Saturated fatty acids play a negative role in long-term cognitive outcomes in stroke patients. The metabolic cascade of polyunsaturated fatty acids (n3 PUFA) and the synthesis of (AA) can be involved in pathogenesis of stroke-related cognitive impairment.

## 1. Introduction

Ischemic stroke is the main cause of physical disability in adults and as much as half of stroke survivors also suffer from cognitive impairments [1]. Although this type of disability is not as noticeable as the motor handicap, it can have a more devastating impact on social functioning than the most frequent motor symptoms, such as hemiparesis. The post-stroke cognitive disability seriously affects quality of life [2]. Moreover, even mild to moderate severity stroke leads to cognitive impairment three months after stroke [3]. Over 60% of stroke survivors will have some degree of cognitive dysfunction, and a third will develop dementia. Conversely, on autopsy, over a third of dementia cases have significant vascular pathology. Vascular cognitive impairment can occur alone or in association with Alzheimer’s disease (AD), with the most severe cognitive impairment observed in those with both pathologies. Anatomic brain changes in vascular cognitive impairment include parenchymal lesions (infarcts and haemorrhages in particular) and the vascular lesions from which they arise [4]. One of the available studies observed differences in plasma and brain fatty acid profiles between patients with AD, mild cognitive impairment (MCI), and those with no cognitive impairment [5].

The fatty acids status has already been investigated and proven to be related to the cognitive functions in healthy subjects [6]. However, there is limited information regarding the role of FFAs in the cognitive outcome in stroke patients. In the neuropathological examination unsaturated fatty acid metabolism dysregulation was observed in the brains of patients with varying degrees of Alzheimer pathology. This referred to such FFAs as linoleic acid, linolenic acid, docosahexaenoic acid, eicosapentaenoic acid, oleic acid, and arachidonic acid [7]. In the Framingham Heart Study, the top quartile of plasma DHA and fish intake was associated with a reduction of all-cause dementia and development of AD during 9.1 years of follow-up period [8]. On the other hand, the fatty fish intake was not found to be connected with the cognitive performance in the six-year observation [9]. Other authors presented the results of an observation with a follow-up period of 35 years. They showed that there was no association between n3 FFAs and the risk of AD, but patients with higher proportions of saturated fatty acids (SFA) were at a lower risk of developing AD [10]. No association between n3 PUFAs and AD was also observed by other authors [11]. In a study where FFAs intake was assessed with the use of nutritional questionnaire, the authors indicated a lower risk of AD with an increasing intake of n3 PUFAs such as DHA and EPA [12]. Dietary intake of n-6 fatty acids was connected with better cognitive functions, but such an observation was not elucidated for other FFAs [13]. Higher levels of omega-6 FFAs consumption including AA were reported to be associated with an increased risk of dementia [14]. Such contradictory and ambiguous observations can be explained by cognitive dysfunction being a complex, chronic, and multifactorial process, especially in the most common types of dementia connected with neurodegenerative pathology, vascular origin or both.

An important public health problem such as dementia is related to another significant component of public health research, namely human nutrition. The appropriate level of fatty acids results mainly from proper nutrition. To date, a number of studies have analysed the impact of FFAs on the pathogenesis and risk of stroke, but there are limited data available regarding the role of FFAs in the pathogenesis of cognitive dysfunctions in stroke patients. The aim of the study was to analyse a potential link between the nutritional influence reflected by the distribution of FFAs with the cognitive outcome and prognosis in the follow-up. To our knowledge, this is the first study to have investigated the associations between certain FFAs and the cognitive functions assessed with neuropsychological tools in patients seven days after the stroke onset and six months later.

## 2. Material and Methods

The protocol of the study was approved by the Ethics Committee in Zielona Góra (decision number 08/73/2017, 28 February 2017). The study was conducted in accordance with the Declaration of Helsinki. We obtained a written, informed consent from our patients to participate in the study.

### 2.1. Subjects

A prospective study was conducted with the participation of 72 ischemic stroke patients who were included according to the inclusion criterion i.e., the diagnosis of the ischemic stroke established on the basis of clinical symptoms and additional tests results, including brain imaging (CT or MRI scans), and who received treatment in accordance with standard protocols and guidelines [15,16]. Eventually, patients with both embolic and atherothrombotic pathomechanism of stroke were included. The ischemic stroke was defined as a syndrome of rapidly developing symptoms of focal or global cerebral dysfunction lasting ≥24 h [17]. The exclusion criteria included intracranial haemorrhage visible in brain imaging, symptoms of active infection including body temperature over 37.4 °C, clinical or biochemical symptoms of infection, an active autoimmune disorder or malignancy as well as speech or consciousness impairment due to cerebral, metabolic or other causes, to exclude unreliable results of neuropsychological tests. Patients with previously diagnosed dementia of any cause were excluded from the study and to minimise the risk of inclusion of a patient with previous dementia, the result in MOCA test of <17 points was also added as the exclusion criterion. Patients were metabolically stabilised when the neuropsychological examination was performed. The initial neurological deficit according to NIHSS score was 4.96 (min. 2, max. 14). The stroke aetiology was classified according to the TOAST classification system. The TOAST classification describes stroke subtypes as follows: large-artery atherosclerosis (*n* = 24), cardioembolism (*n* = 9), small vessel occlusion/lacunar (*n* = 25), other determined causes (*n* = 0), and undetermined cause (*n* = 14) [18]. All subjects had FFAs gas chromatography performed. The patients were hospitalised in the Neurology Department in the district hospital in Poland. All patients were evaluated by the neuropsychologist with the use of tests on the seventh day after stroke. The follow-up neuropsychological examination was performed after 6 months (*n* = 41). The reduced number of patients in the follow-up resulted from their failure to report for a follow-up visit. All patients were Caucasians. None of the patients was taking omega-3 supplementation before admission to hospital. During hospitalisation, all patients were treated with statins and acetylsalicylic acid. We also analysed potential confounding effects of certain comorbidities and lipid profile (total cholesterol, HDL-C, LDL-C, triglycerides and non-HDL cholesterol) on FFAs and neuropsychological tests. Detailed characteristics of the study group was presented in Table 1.

We presented mean values of free fatty acids in the analysed group of stroke patients and controls (*n* = 30). The control group consisted of stroke-free adults aged 63.1 years (females *n* = 18). There were no significant differences identified between stroke and control groups regarding age and gender.

### 2.2. Neuropsychological Assessment

The neuropsychological assessments were performed by a qualified neuropsychologist experienced in diagnosing cognitive dysfunctions and stroke. Several tests were selected to evaluate the cognitive performance in stroke patients. The California Verbal Learning Test 2nd edition was used to assess memory, learning, recognition and attention. Selected tasks were analysed: List A, List B, short delay free recall (SDFR), short delay cued recall (SDCR), long delay free recall (LDFR), long delay cued recall (LDCR) and semantic clustering [19,20]. The results were given as sten values. STEN scores is a psychological test scale normalised so that the population mean is 5.5 and the standard deviation is 2. The scale is divided into 10 units. Interpretations of the results of psychological tests refer to standardised scales, the interpretation of raw results requires the conversion of the result obtained in the test into the result on a normalised scale. The sten scale is a standardised scale for converting the obtained raw results in the CVLT test for all test indices and is commonly used in clinical practice by neuropsychologists. The use of this scale is helpful in correlation assessment which enables comparisons and a more precise interpretation of the obtained results. Higher results indicated better cognitive outcome.

Trail Making Test consisted of two parts (A and B) and was used for the assessment of psychomotor velocity, concentration, visuospatial attention and search, working memory, mental flexibility, and ability to task switch. The score is represented by the time in seconds. Higher results indicated worse cognitive outcome [21,22].

Forward and Backward Digit Span Test (FDST and BDST) is a subtest of the Wechsler Adult Intelligence Scale and the Wechsler Memory Scale and was used to assess the short-term verbal memory, efficiency and capacity of attention and verbal working memory [23,24].

The Stroop Color and Word Test (SCWT) also known as the Stroop Dots Trial A and B (SDT A and SDT B) was used to assess the ability to inhibit cognitive interference and to assess the plastic adaptation to the changing rules of the task [25]. We used Polish, experimental version which was adapted from the original test. The Verbal Fluency Test was performed in two parts (category fluency and letter fluency). The category of fluency was composed of the following parts: Verbal Fluency Test-Animals (VFT-A), Verbal Fluency Test-Fruits and Vegetables (VFT-F&V) and letter fluency was evaluated in: Verbal Fluency Test-Letter “K” (VFT-K) and Verbal Fluency Test-Letter “P” (VFT-P). We used Polish version which was adapted from the original test. The VFT was applied to evaluate executive function, expressive language ability, memory and processing speed [22]. All neuropsychological tests marked with “2” indicate results of tests performed after 6 months. Higher scores in all but TMT and SDT tests indicated better cognitive outcomes in certain aspects of cognitive functions.

The Montreal Cognitive Assessment (MOCA) test was selected as a simple and clinically useful tool for the assessment of the main cognitive domains such as attention, memory, orientation, language, visuospatial ability and abstract thinking [26]. All patients were divided into two subgroups according to their MOCA score and then their FFAs levels were compared. Subgroup I included patients with MOCA test result <26 (group I, *n* = 35), while subgroup II consisted of patients with MOCA score ≥26 points (group II, *n* = 37). This test was performed only on the 7th day of our study.

### 2.3. Blood Collection and Fatty Acids Detection

Venous blood samples were collected seven days after the onset of the symptoms (*n* = 72). The analyses of the FFAs with the use of liquid and gas chromatography (Agilent Technologies 7890A GC System equipped with a SUPELCOWAX 10 Capillary GC Column) were performed after centrifugation and stored at −80 °C. The methyl esters of FFAs were isolated from serum with the use of the modified Folch and Szczuko methods [27,28].

Detailed methodology of free fatty acids detection was described elsewhere [29]. The results are presented as the percentage of the individual fatty acids in the total mass of fatty acids in the examined samples. The following FFAs were identified in the samples: C13:0 tridecanoic acid, C14:0 myristic acid, C14:1 myristolenic acid, C15:0 pentadecanoid acid, C15:1 cis-10-pentadecanoid acid, C16:0 palmitic acid, C16:1 palmitoleic acid, C17:0 heptadecanoic acid, C17:1 cis-10- heptadecanoid acid, C18:0 stearic acid, C18:1n9 ct oleic acid, C18:1 vaccinic acid, C18:2n6c linoleic acid, C18:2n6t linoleic acid, C18:3n6 gamma linoleic acid, C18:3n3 linolenic acid, C18:4 stearidonic acid, C20:0 arachidic acid, C22:1/C20:1 Cis11- eicosanic acid, C20:2 Cis-11-eicodienoic acid, C20:3n6 eicosatrienoic acid, C20:4n6 arachidonic acid, C20:3n3 Cis-11-eicosatrienoic acid, C20:5n3 eicosapentaenoic acid, C22:0 behenic acid, C22:1n9 13 erucic acid, C22:2 cis-docodienoic acid, C23:0 tricosanoic acid, C22:4n6 docosatetraenoate, C22:5w3 docosapentaenate, C24:0 lignoceric acid, C22:6n3 docosahexaenoic acid, C24:1 nervonic acid.

### 2.4. Statistical Analysis

The means and statistical deviations were calculated for both MOCA subgroups. To assess the equality of variances for variables the Levene’s test was used before a comparison of means. The test has reached the level of significance (*p* < 0,05). For this reason and because of non-normality of the distributions between variables (Shapiro–Wilk test), the numerical data were compared between groups using the nonparametric Mann–Whitney U-test. For two groups of repeated variables Wilcoxon’s test was used. The occurrence of nominative clinical data was compared by means of chi-squared test if needed. A potential impact of comorbidities on FFAs or neuropsychological tests was tested with the use of chi^2^ Pearson’s correlation for quantitative variables, while Spearman’s rank correlation was used for nominative variables. The correlation matrix was obtained for FFAs levels and the results of neuropsychological tests (CVLT, TMT, FDST, BDST, SDT, VFT) recorded on the 7th day after stroke and 6 months later. *p* < 0.05 was considered to indicate statistical significance. Statistical analyses were performed with Statistica 13 (Statsoft, Cracow, Poland).

## 3. Results

Table 2 and Table 3 present the statistically significant associations between free fatty acids measured on 7th day and results of CVLT test performed on 7th day post-stroke and after 6 months respectively. The scores are given as sten values. First, the associations between the levels of FFAs and CVLT test results obtained on the 7th day after stroke were analysed (Table 2). We observed significant direct correlations regarding AA and EPA. The negative correlations were detected in ALA, C18:4 n3 stearidonic acid (SDA) and C23:0 tricosanoic acid. Linolenic acid, EPA and C20:4n6 arachidonic acid were significantly associated with one or two tasks of CVLT test, while SDA and C23:0 tricosanoic acid were positively associated with most parts of the analysed test. We did not find any significant associations with regard to List B of CVLT and any of the analysed FFAs. nor did we find any significant correlations regarding other analysed FFAs on the 7th day.

The analysis of correlations between FFAs and CVLT test results in the follow-up after 6 months is presented in Table 3. Four fatty acids including C18:3n6 gamma linoleic acid (GLA) and SDA were directly correlated with LDCR task. C14:0 myristic acid and C14:1 myristolenic acids were negatively correlated with list A task of CVLT test, while ALA was negatively associated with three tasks of CVLT test. We did not detect any significant associations with regard to List B and semantic clustering of CVLT and any of FFAs. No significant correlations were observed regarding other analysed FFAs and CVLT test performed after 6 months.

Figure 1 presents a cascade of C18:3n3 linolenic acid (LA) metabolism to clarify the stages that are correlated with the results of CVLT. When taken together, the results obtained in the initial and the follow-up neuropsychological assessments indicated consistent associations in metabolic sequences of LA, GLA, and AA.

Table 4 shows the correlations between FFAs and certain neuropsychological tests results obtained 7 days after stroke: Trail Making Test (TMT) tasks A and B, Stroop Dots Trail tasks A and B (SDT A and SDT B), Digit Span Test (DST) and Verbal Fluency Test (VFT). The results of the majority of these tests were directly associated with C14:0 myristic acid and C14:1 myristolenic acid. TMT test (task A) and Stroop Dots Trail test (task A) results were correlated with only one fatty acid i.e., C18:1 vaccinic acid (direct association). Digit Span Test (forward and backward) results were directly associated with 14:0 myristic acid, C14:1 myristolenic acid, C15:0 pentadecanoid acid, C17:0 heptadecanoic acid, GLA, while negative associations were observed in C18:1 vaccinic acid and C20:3n3 cis-11-eicosatrienoic acid. Several tasks in the assessment of verbal fluency were directly correlated with C14:0 myristic acid, C14:1 myristolenic acid and only in individual tasks directly associated with C16:0 palmitic acid, C16:1 palmitoleic acid, C17:0 heptadecanoic acid, and negatively correlated with C22:1/C20:1 cis11-eicosanic acid and C20:2 cis-11-eicodienoic acid. TMT task A results were directly and significantly correlated only with C18:1 vaccinic acid. We did not observe any significant associations with regard to VFT-P and any of FFAs. Neither did we identify any significant correlations regarding other analysed FFAs on the 7th day.

In the follow-up examination after 6 months, the study participants were individually and repeatedly assessed with the use of neuropsychological tests (Table 5). C18:2n6t Linoleic acid was directly associated with the outcomes of SDT and two tasks of VFT. Several FFAs were associated only with individual results while C14:0 myristic acid, C14:1 myristolenic acid and C15:0 pentadecanoid acid were directly associated with the performance in DST and VFT, and task A of SDT. C20:0 arachidic acid and C22:5w3 docosapentaenate were negatively correlated with SDT results. We did not find any significant associations with regard to TMT A, VFT-P and BDST and any of FFAs. We did not detect any significant correlations either regarding other analysed FFAs and tests presented in Table 5 performed after 6 months.

The results of TMT B and SDT B tests performed after 7 days as well as after 6 months did not show significant associations with the levels of analysed free fatty acids. The correlations between FFAs and neuropsychological test scores performed on the 7th day and after 6 months are presented in the graphs as the Figure 2 and Figure 3 respectively.

No correlations between MOCA test and FFAs levels were found. We additionally compared patients with MOCA test result <26 (group I, *n* = 35, mean result 22.4, min. 17, max. 25) versus ≥26 points (group II, *n* = 37, mean result 27.8, min. 26, max. 30) regarding the levels of free fatty acids. However, no statistically significant differences were identified between the two groups. We only detected a trend in the level of C18:1n9 ct oleic acid towards higher values in patients with cognitive impairment (*p* = 0.058, group I: mean ± SD 23.4 ± 3.87, group II: mean ± SD 21.81 ± 3.42).

We additionally correlated comorbidities with FFAs. We found significant negative correlations between the presence of hypertension and C18:1 vaccinic acid and C18:3n3 linolenic acid (Spearman’s rank correlation r = −0.902 and r = −0.942, respectively), diabetes was correlated with C14:0 myristic acid (r = −0.816) and C22:2 cis-docodienoic acid (r = 0.982). Coronary heart disease, thyroid disease (hypothyroidism), smoking and alcohol consumption were not found to be correlated with FFAs. The age was negatively correlated with C22:1n9 13 erucic acid (chi^2^ Pearson’s correlation r = −0.819).

Table 6 presents the results of neuropsychological tests after 7 days and 6 months later.

In Table 7, there is presented a comparison of FFAs between stroke and control group.

The analysis between certain demographic, nutritional factors, comorbidities and neuropsychological is presented in Table 8 (only statistically significant results are presented). Gender, hypothyroidysm and other lipid profile parameters (LDL, total cholesterol and non-HDL presented in mg/dl) were not correlated with any of the neuropsychological tests.

## 4. Discussion

As the first step of our outcome assessment, we analysed the association between FFAs and the results of CVLT test performed on the 7th day after stroke (Table 2).

Inverse associations were found between ALA and cognitive performance in certain parts of the CVLT performed both 7 days and 6 months after stroke. This indicates that higher ALA level assessed 7 days after stroke is associated with worse cognitive outcomes at that time and in a longer period of observation, while EPA could be treated as a positive predictor. C18:3n3 α-linolenic acid and C20:5n3 eicosapentaenoic acid are the n3 essential, polyunsaturated fatty acid that constitutes an important part of the Mediterranean diet, which has been proven to decrease the risk of cardiovascular disorders including stroke, mild cognitive impairment and neurodegenerative diseases. Being connected with vascular pathology, the cerebral white matter hyperintensities (WMH) correlate with the risk of cerebrovascular episodes and dementia. DHA level is significantly lower in MCI and AD patients, and lower serum EPA/AA ratio is associated with the progression of the white matter hyperintensities [30,31]. Dietary DHA and its metabolites may decrease amyloid-β apoptosis and oxidative stress that may have pathogenetic importance in AD patients [32]. Moreover, higher levels of DHA are connected with a decreased risk of all-cause dementia [8]. In a small sample size of 13 subjects, the n3 intake was reported to improve cerebral perfusion in patients with mild cognitive impairment [33]. The cognitive performance in the elderly patients was also found to be positively associated with the total n3 PUFAs by Baierle M. et al. but the CVLT was not implemented in their study [6]. On the other hand, n3 fatty acids intake was not proven to affect the incidence of dementia in the meta-analysis [34,35]. In the analysis of the brain tissue of AD patients, DHA was significantly elevated in comparison to healthy brains [36]. The role of n3 PUFAs in cognitive disorders may result from a further conversion to eicosanoids—the anti-inflammatory lipid mediators. The role of inflammatory metabolites of n3 PUFAs may be observed in the pathogenesis of both vascular and degenerative cognitive disorders [37]. Our study results which indicate a negative correlation of ALA with cognitive functions could be interpreted regarding further interesting findings (Figure 1)—C18:3n6 gamma linoleic acid was positively correlated with the cognitive outcome in the CVLT after six months. ALA is a substrate for GLA which is subsequently elongated to C20:3n6 eicosatrienoic acid and then to C20:4n6 arachidonic acid [38]. It can be suggested that lower levels of ALA may be a consequence of GLA synthesis, which may affect further inflammatory cascade within the inflammatory metabolites of AA i.e., eicosanoids. Such a presumption was also confirmed by the increased AA synthesis observed in our study. In one animal study, ALA supplementation was shown to improve hippocampal neurons survival, memory and spatial learning after ischemic stroke [39]. Otherwise, there is limited information in the available literature regarding the role of ALA and EPA in cognitive outcomes in stroke patients.

Beside the CVLT, other neuropsychological tests were also applied on the seventh day and six months after stroke. Our study subjects were neuropsychologically assessed with the use of Trail Making Test tasks A and B, Stroop Dots Trail tasks A and B, Digit Span Test and Verbal Fluency Test. Higher scores in all but TMT and SDT tests indicated better cognitive outcomes in certain aspects of cognitive functions. The results of most of these tests were directly associated with C14:0 myristic acid and C14:1 myristolenic acid, and these acids seemed to have the strongest direct relationship with the cognitive performance in stroke survivors but only in the initial assessments. The associations between SFAs and follow-up tests results (CVLT and SDT A) indicate that these acids are negative predictors of cognition in the longer run. In the available literature SFAs were mainly reported as positively associated with AD, total dementia, MCI and cognitive decline, while their inverse relationship with AD, although less documented, was still recorded as significant [40]. SFA-rich diet can disrupt the blood-brain-barrier (BBB) in the animal model, which can disrupt synapses, increase the development of amyloid angiopathy and impair cognition [41]. So, the follow-up tests results in our study are consistent with the outcomes observed by other authors. The only differences between the available data and our results were detected in the initial examination and may be connected with the fact that previous studies focused mainly on AD with no analyses of the acute phase of stroke, while there are differences in the long-term pathogenesis of neurodegenerative disorders and stroke. Moreover, in this study cognitive functions were examined early after stroke and in the follow-up setting, which had not been examined to such extent before and which allowed us to observe interesting relationships. Other authors showed that SFAs, including C14:0 myristic acid, were inversely correlated with the cognitive performance in the elderly patients but their analysis included only 45 subjects. The cognition was assessed with the use of several neuropsychological tests including Verbal Fluency, Word List Memory, Trail Making Test, however, the CVLT was not applied in their study [6]. A potential role of SFAs in cognitive function was shown in the AD: palmitic acid was recorded to be involved in the amyloid plaque formation by triggering the production of Aβ peptide [42]. SFAs increase the level of LDL-C in the blood and may lead to the development of atherosclerosis. Increased severity of atherosclerosis may contribute to vascular dementia which is one of the factors responsible for the neuropsychological status. The positive association of SFAs with cognition in our study in the early phase of stroke, but negative relationships in the follow-up examination, may potentially differ pathogenetically as we can observe different pathogenetic influence in the acute phase of stroke and over a longer period. Our findings may also differ from the outcomes obtained in non-stroke patients. On the other hand, we detected a negative role of SFAs in cognitive performance in the follow-up examination with CVLT and SDT A after six months, which is consistent with the findings cited above. Some shortcomings of this study should be acknowledged. Firstly, our study did not include patient differentiation according to the pathogenetic background of their cognitive deficits. A detailed neuropsychological diagnosis could be included in the design to address this limitation. Moreover, the neuropsychological assessments performed on the seventh day post-stroke could be inaccurate as they might be interpreted as an acute brain damage. The analysis after six months of follow-up period, however, is undoubtedly the study advantage.

C18:1 vaccinic acid was the only fatty acid which showed direct association with TMT A and SDT A results thereby indicating worse psychomotor activity, concentration and visuo-spatial search. This FFA was found to be negatively associated with backwards repeating in Digit Span Test, which suggests its ambiguous role in predicting the working memory. In a limited number of studies available, no differences were identified in serum vaccinic acid level between AD patients and controls [43]. There are no available data regarding vaccinic acid and cognitive outcome in stroke patients.

In addition to the above-mentioned results, the neuropsychological assessment was carried out again after six months. In the follow-up examination, a direct relationship was found between C18:2n6t linoleic acid and the results of Digit Span Test and two tasks of Verbal Fluency Test, which suggests its prominent and positive role in preserving the cognitive functions in a longer time span. The tests outcomes indicated better access to working and long-term memory, enhanced efficiency in extracting information from long-term memory and increased efficiency of executive functions and memory searching strategies. This FA belongs to the PUFA n6 group that are supposed to be negatively connected with the cognitive functions, however, the opposite activity was also observed by other authors [13,14]. Alas, such information regarding stroke patients is yet to be confirmed and published. More of the available evidence indicates the role of n6 to n3 PUFA ratio in the cognitive dysfunctions [44,45]. Other studies did not report any differences of this FA in AD patients compared to controls [43]. Observations presented by other authors differ from our results, but it should be emphasised that we investigated a specific group of stroke survivors and similar analyses have not been presented yet. The other analysed FFAs were associated only with individual results in the follow-up examination, but three of them should be mentioned as they were also significantly associated with the neuropsychological tests in the first assessment, seven days post-stroke. C14:0 myristic acid, C14:1 myristolenic acid and C15:0 pentadecanoid acid were directly associated with the results of Digit Span Test and Verbal Fluency assessments made on the seventh day and task A of Stroop Dots Trial carried out after six months. On the other hand, in the follow-up examination, C20:0 arachidic acid and C22:5w3 docosapentaenate were negatively correlated with Stroop Dots Trial, which indicates the beneficial influence on cognitive outcome and their potentially positive role in promoting neuroplasticity. The role of SFAs in the cognitive dysfunctions presented by other authors was discussed above.

It is also noteworthy that C24:1 nervonic acid was not related to any aspects of cognitive functions in our patients at any of the assessment points. This fatty acid was demonstrated to be an important part of biosynthesis of myelin. The oil rich in nervonic acid increases the synthesis of proteolipid protein, sphingomyelin, myelin basic protein and myelin oligodendrocyte glycoprotein [46].

There can also be a confounding effect of comorbidities on FFAs and neuropsychological tests, which was presented in Table 8. This especially refers to age, CHD, hypertension, BMI, and diabetes that are associated with nutritional background in patients and are potential risk factors of vascular dementia.

We detected slightly lower levels of C18:1n9 ct oleic acid in the subjects with normal results in MOCA test compared to the other patients (*p* = 0.058). There is limited information regarding its role in the cognitive functions in humans. In the analysis of FFAs distribution in the brain tissue of AD patients the level of C18:1n9 ct oleic acid was recorded to remain unaltered [35]. MOCA test is used as a clinical screening tool but it does not indicate particular cognitive deficits as opposed to the other tests applied in our study.

The percentage of fatty acids was compared between the study group and the control group (Table 7). In the studied group of patients with stroke, higher concentrations of acids such as C18: 1 vaccinic acid, C16: 0 palmitic acid, GLA, C18: 3n6 gamma linoleic acid were observed, while lower concentrations of ALA or C20: 3n3 cis-11-eicosatrienoic acid. In this study, however, we focused on the analysis of fatty acids in relation to cognitive functions. The relationship between fatty acids between patients with stroke and the control group indicates the importance of the metabolic pathways of FFAs in stroke patients and requires further research [29].

On the basis of the obtained results, we can suggest that a diet rich in n6 PUFA and n3 PUFA may have a positive effect on cognitive functions in patients after stroke. This is especially true of consuming seeds and oils such as flaxseed oil, chia seeds, green soybeans and soybean oil, avocado, whole wheat products, oatmeal or nuts, especially walnuts. An additional dietary factor that positively influences cognitive functions in this group of patients seems to be limiting the consumption of SFAs. It particularly applies to products such as whole-fat dairy products or red meat. Our findings in some aspects confirm the beneficial effects of MIND (Mediterranean-DASH Intervention for Neurodegenerative Delay) diet, but on the biochemical level. This diet was proved to have beneficial effects in cognitive decline and decreases the risk of developing dementia [47].

Several limitations of our study should be acknowledged and they include a lack of relation to neuroimaging and the size of infarction as we were chiefly concentrated on the assessment of the cognitive functions. Next, we had no information regarding the factors that could affect patients’ cognitive performance during the follow-up period. Another limitation refers to the follow-up neuropsychological examination which was performed only in 41 patients out of the initial group of 72, which resulted from the fact that patients did not respond to our invitation to the follow-up neuropsychological examination. We are aware of the low number of participants, so our conclusions should be interpreted as a novel, pilot study.

## 5. Conclusions

We demonstrated several significant associations between certain FFAs and cognitive functions in stroke survivors and we discussed cognitive changes detectable in the follow-up examination six months after the acute phase of stroke. In clinical practice, post-stroke dementia should not be diagnosed directly after stroke. Therefore, in our study, the neuropsychological assessment was carried out six months post-stroke, which allowed for the exclusion of the potential impact of confounders resulting from the acute brain damage. However, FFAs are involved both in the stroke pathogenesis and in the cognitive functions so their role in the neuropsychological aspects of these two parallel processes is also interesting in the acute phase of stroke. Moreover, the long-term cognitive outcome is related to the acute-phase cognitive status and has been documented to be affected by FFAs in both periods. In addition, the same FFAs may exert diverse effects in the acute phase and in the long-term observation, which was detected regarding SFAs. Therefore, the role of FFAs in the acute phase of stroke cognition also needs further studies. Future investigations should also take into consideration any potential associations between FFAs, their inflammatory metabolites and the cognitive performance in stroke survivors.

We suggest that FFAs and thus dietary aspects can have an impact on the development of dementia in a complex and chronic process of multifactorial interactions. This especially refers to a diet rich in n6 PUFA, n3 PUFA, and limiting the consumption of SFAs. The role of FFAs in the pathogenesis of dementia may result from direct effects on lipid metabolism, the impact on atherosclerosis or interactions with inflammatory mediators. Fatty acids are the substrates in the metabolic pathways of the inflammatory derivatives. The n-3 fatty acids are metabolised into the lipid mediators through the cyclooxygenase (COX) and lipooxygenase (LOX) pathways. The specialised pro-resolving lipid mediators (SPMs) have anti-inflammatory properties and can be involved in the maintenance of the neuroinflammatory status and may affect the cognitive functions. EPA gives rise to such SPM as resolvin E series including resolvin E1, E2, and E3, while DHA is converted to maresins, protectins, and resolvin D-series [48]. On the other hand, most of the eicosanoids formed from arachidonic acid such as prostaglandin E2, leukotriene B4, and thromboxane A2 are pro-inflammatory, whereas lipoxins have anti-inflammatory, pro-resolving properties [29,49]. Such complex interactions of these metabolic pathways need further studies regarding the impact of FFAs on cognition in stroke patients.

## Figures and Tables

**Figure 1 ijerph-18-06500-f001:**
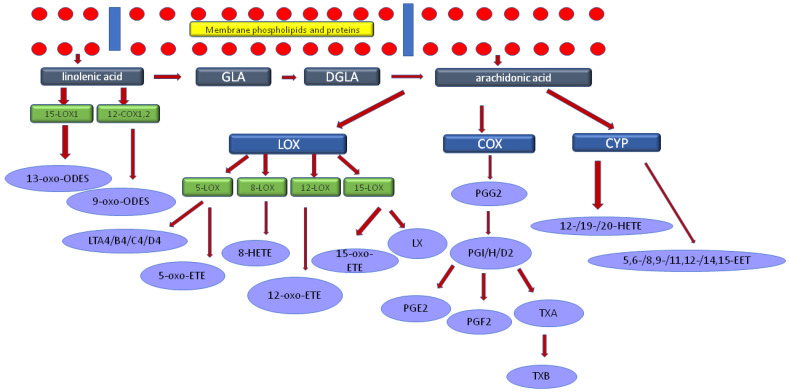
The cascade of linolenic acid metabolism. Legend: GLA-C18:3n6 gamma linoleic acid, DGLA-dihomo-γ-linolenic acid (C20:3n6 eicosatrienoic acid), LOX-lipoxygenase, COX-cyclooxygenase, CYP-cytochrome, LT-leukotriene, LX-lipoxin, PG-prostaglandins, TX-thromboxan. Linolenic acid was negatively correlated with the results of CVLT test, while GLA and AA were directly associated.

**Figure 2 ijerph-18-06500-f002:**
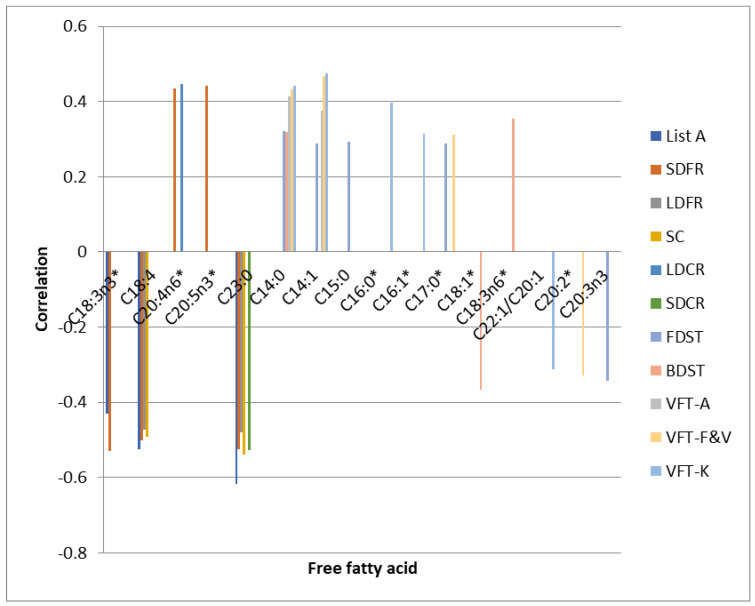
Correlations between FFAs and neuropsychological test scores performed on the 7th day. * statistically significant differences between the FFA level in the control and experimental group.

**Figure 3 ijerph-18-06500-f003:**
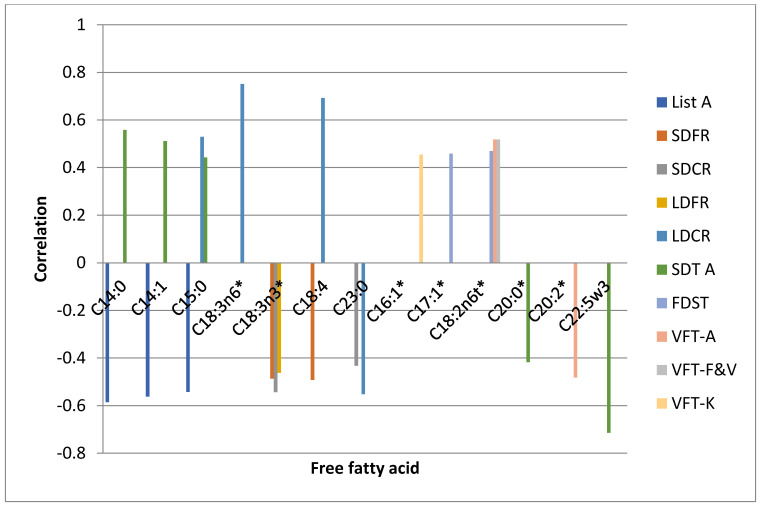
Correlations between FFAs and neuropsychological test scores performed after 6 months. ***** statistically significant differences between the FFA level in the control and experimental group.

**Table 1 ijerph-18-06500-t001:** Characteristics of the study group (*n* = 72).

Parameter	Result	Parameter	Result
Sex (female)	40 (55.5%)	Hypothyroidism	11 (15.3%)
BMI (kg/m^2^)	28.94 (min. 17.7, max. 46.2)	Obesity (BMI ≥25 kg/m^2^)	59 (81.9%)
Age (years)	60.7 (min. 25, max. 83)	Hypertension	61 (84.7%)
Diabetes/impaired fasting glucose (n)	36 (50%)	Intravenous alteplase infusion	7 (9.7%)
Excessive drinking	8 (11.1%)	Hypoglycaemic drugs before hospitalisation	17 (23.6%)
Smoking	26 (36.1%)	Hypotensive agents before hospitalisation	61 (84.7%)
Coronary heart disease	8 (11.1%)	L-thyroxine before hospitalisation	11 (15.3%)

**Table 2 ijerph-18-06500-t002:** The correlations between free fatty acids (FFAs) and Cognitive Verbal Learning Test (CVLT) score obtained on 7th day after stroke (CVLT 1).

FFA [%]	List A [sten]	SDFR [sten]	SDCR [sten]	LDFR [sten]	LDCR [sten]	SC [sten]
C18:3n3 linolenic acid	−0.444 *	−0.236	−0.132	−0.294	−0.282	−0.124
C18:4 stearidonic acid	−0.524 *	−0.502 *	−0.209	−0.473 *	−0.408	−0.492 *
C20:4n6 arachidonic acid	0.114	0.436 *	0.419	0.16	0.446*	−0.091
C20:5n3 eicosapentaenoic acid	0.283	0.443 *	0.252	0.23	0.221	0.139
C23:0 tricosanoic acid	−0.618 *	−0.525 *	−0.528 *	−0.481 *	−0.402	−0.538 *

* *p* < 0.05 statistically significant correlation matrix (Spearman’s rank correlation); SDFR—short delay free recall, SDCR—short delay cued recall, LDFR—long delay free recall, LDCR—long delay cued recall, SC—semantic clustering.

**Table 3 ijerph-18-06500-t003:** The correlations between FFAs and CVLT score obtained 6 months after stroke (CVLT 2).

FFA [%]	List A [sten]	SDFR [sten]	SDCR [sten]	LDFR [sten]	LDCR [sten]
C14:0 myristic acid	−0.585 *	−0.373	−0.111	−0.18	0.3
C14:1 myristolenic acid	−0.562 *	−0.271	0.017	−0.011	0.419
C15:0 pentadecanoid acid	−0.542 *	−0.324	−0.078	−0.182	0.529 *
C18:3n6 gamma linoleic acid	−0.122	−0.239	0.053	−0.045	0.751 *
C18:3n3 linolenic acid	−0.321	−0.486 *	−0.543 *	−0.462 *	0.31
C18:4 stearidonic acid	−0.408	−0.492 *	−0.209	−0.299	0.692 *
C23:0 tricosanoic acid	−0.267	−0.276	−0.432 *	−0.326	0.552 *

* *p* < 0.05 statistically significant correlation matrix (Spearman’s rank correlation).

**Table 4 ijerph-18-06500-t004:** The correlations between FFAs and results of neuropsychological tests performed on the 7th day after stroke (TMT A, SDT A, FDST, BDST, VFT-A, VFT-F&V, VFT-K, VFT-P).

FFA [%]	TMT A	SDT A	FDST	BDST	VFT-A	VFT-F&V	VFT-K
C14:0 myristic acid	0.01	−0.13	0.322 *	0.318 *	0.414	0.433 *	0.441 *
C14:1 myristolenic acid	0.124	−0.161	0.288 *	0.213	0.376 *	0.469 *	0.474 *
C15:0 pentadecanoid acid	−0.018	0.183	0.293 *	0.042	0.123	0.152	0.240
C16:0 palmitic acid	−0.132	−0.118	0.167	0.188	0.245	0.241	0.398 *
C16:1 palmitoleic acid	0.205	−0.076	−0.077	0.108	0.063	0.097	0.315 *
C17:0 heptadecanoic acid	−0.227	−0.082	0.289 *	0.194	0.242	0.311 *	0.193
C18:1 vaccinic acid	0.327 *	0.352 *	−0.212	−0.366 *	−0.111	0.02	−0.013
C18:3n6 gamma linoleic acid	−0.013	−0.202	0.208	0.355 *	0.068	0.061	0.17
C22:1/C20:1 Cis11- eicosanic acid	0.007	0.052	0.013	−0.154	−0.084	−0.105	−0.312 *
C20:2 Cis−11-eicodienoic acid	0.088	0.014	0.029	−0.133	−0.148	−0.329 *	−0.054
C20:3n3 Cis-11-eicosatrienoic acid	0.223	0.132	−0.342 *	−0.269	−0.025	−0.089	−0.075

* *p* < 0.05 statistically significant correlation matrix (Spearman’s rank correlation); TMT A—Trial Making Test A, SDT A—Stroop Dots Trial A, FDST—Forward Digit Span Test, BDST—Backward Digit Span Test, VFT-A—Verbal Fluency Test-Animals, VFT-F&V—Verbal Fluency Test-Fruits and Vegetables, VFT-K–Verbal Fluency Test-Letter “K”.

**Table 5 ijerph-18-06500-t005:** The correlations between FFAs and the results of neuropsychological tests obtained after 6 months (SDT A, FDST, BDST, VFT-A, VFT-F&V, VFT-K, VFT-P).

FFA [%]	SDT A	FDST	BDST	VFT-A	VFT-F&V	VFT-K
C14:0 myristic acid	0.558 *	−0.215	0.051	−0.326	−0.105	−0.03
C14:1 myristolenic acid	0.511 *	−0.285	0.029	−0.264	−0.054	−0.004
C15:0 pentadecanoid acid	0.442 *	−0.324	−0.27	−0.083	0.068	−0.033
C16:1 palmitoleic acid	0.17	−0.022	0.076	0.095	0.024	0.454 *
C17:1 cis−10- heptadecanoid acid	0.118	0.458 *	−0.283	0.139	0.23	−0.187
C18:2n6t linoleic acid	−0.06	0.469 *	0.166	0.518 *	0.518 *	0.323
C20:0 arachidic acid	−0.417 *	−0.203	0.286	−0.045	0.201	0.025
C20:2 Cis−11-eicodienoic acid	0.336	0.104	0.202	−0.482 *	−0.297	−0.142
C22:5w3 docosapentaenate	−0.714 *	−0.07	−0.073	0.006	−0.194	0.024

* *p* < 0.05 statistically significant correlation matrix (Spearman’s rank correlation); SDT A—Stroop Dots Trial A, FDST—Forward Digit Span Test, VFT-A—Verbal Fluency Test-Animals, VFT-F&V—Verbal Fluency Test-Fruits and Vegetables, VFT-K—Verbal Fluency Test-Letter “K”.

**Table 6 ijerph-18-06500-t006:** Results of neuropsychological tests performed on the 7th day and after 6 months.

Test	Result at 7th Day (Mean, Min., Max.)	Result after 6 Months	Test	Result at 7th Day (Mean, Min., Max.)	Result after 6 Months
List A (CVLT)	5.94 (2, 10)	6.78 (4, 10)	TMT A	54.14 (20, 143)	47.07 (18, 122)
List B (CVLT)	5.87 (2,10)	6.5 (2, 10)	TMT B	136.68 (38, 381)	119.56 (44, 360)
SDFR (CVLT)	5.89 (1, 10)	6.21 (2, 9)	FDST	5.21 (2, 8)	5.45 (2, 8)
SDCR (CVLT)	5.73 (1, 10)	6.36 (2, 10)	BDST	3.79 (1, 11)	4.1 (1, 6)
LDFR (CVLT)	6.1 (1, 10)	6.57 (3, 10)	VFT-A	20.91 (9, 33)	22.23 (9, 34)
LDCR (CVLT	5.6 (1, 10)	6.14 (2, 10)	VFT-F&V	19.37 (10, 31)	19.26 (11, 30)
SC (CVLT)	5.25 (1, 10)	6.11 (1, 10)	VFT-K	9.98 (1, 24)	12.06 (4, 21)
SDT A	96.89 (53, 212)	84.78 (3, 134)	VFT-P	10.06 (2, 22)	10.74 (4, 22)
SDT B	167.1 (91, 412)	139.39 (3, 250)			

**Table 7 ijerph-18-06500-t007:** Comparison between FFAs of stroke patients (*n* = 72) and control group (*n* = 30).

FFA	Stroke Group (Mean ± SD)	Control Group (Mean ± SD)	*p*	FFA	Stroke Group (Mean ± SD)	Control Group (Mean ± SD)	*p*
C13:0 tridecanoic acid	0.305 ± 0.09	0.377 ± 0.12	<0.05	C20:0 arachidic acid	0.206 ± 0.07	0.147 ± 0.04	<0.05
C14:0 myristic acid	1.22 ± 0.38	1.129 ± 0.34	NS	C22:1/C20:1 Cis11- eicosanic acid	0.18 ± 0.07	0.188 ± 0.06	NS
C14:1 myristolenic acid	0.07 ± 0.04	0.069 ± 0.03	NS	C20:2 Cis-11-eicodienoic acid	0.15 ± 0.03	0.17 ± 0.03	<0.05
C15:0 pentadecanoid acid	0.215 ± 0.12	0.214 ± 0.05	NS	C20:3n6 eicosatrienoic acid	1.275 ± 0.3	1.255 ± 0.24	NS
C15:1 cis-10-pentadecanoid acid	0.079 ± 0.03	0.133 ± 0.38	<0.05	C20:4n6 arachidonic acid	6.284 ± 1.33	7.117 ± 1.43	<0.05
C16:0 palmitic acid	26.796 ± 1.77	25.706 ± 1.31	<0.05	C20:3n3 Cis-11-eicosatrienoic acid	0.031 ± 0.01	0.047 ± 0.01	<0.05
C16:1 palmitoleic acid	2.151 ± 0.75	1.715 ± 0.49	<0.05	C20:5n3 eicosapentaenoic acid	0.608 ± 0.26	1.233 ± 0.8	<0.05
C17:0 heptadecanoic acid	0.301 ± 0.05	0.335 ± 0.04	<0.05	C22:0 behenic acid	0.225 ± 0.99	0.087 ± 0.04	<0.05
C17:1 cis-10-heptadecanoid acid	0.09 ± 0.03	0.135 ± 0.04	<0.05	C22:1n9 13 erucic acid	0.037 ± 0.02	0.033 ± 0.01	NS
C18:0 stearic acid	13.258 ± 1.83	14.801 ± 2.01	<0.05	C22:2 cis-docodienoic acid	0.017 ± 0.01	0.025 ± 0.01	<0.05
C18:1n9 ct oleic acid	22.645 ± 3.71	18.412 ± 2.6	<0.05	C23:0 tricosanoic acid	0.235 ± 0.15	0.22 ± 0.39	NS
C18:1 vaccinic acid	1.977 ± 0.35	1.713 ± 0.25	<0.05	C22:4n6 docosatetraenoate	0.216 ± 0.11	0.263 ± 0.13	NS
C18:2n6c linoleic acid	11.53 ± 2.37	12.84 ± 1.89	<0.05	C22:5w3 docosapentaenate	0.464 ± 0.23	0.578 ± 0.09	<0.05
C18:2n6t linoleic acid	6.162 ± 1.89	7.342 ± 1.46	<0.05	C24:0 lignoceric acid	0.154 ± 0.08	0.043 ± 0.04	<0.05
C18:3n6 gamma linoleic acid	0.391 ± 0.19	0.264 ± 0.09	<0.05	C22:6n3 docosahexaenoic acid	1.746 ± 0.54	2.405 ± 0.71	<0.05
C18:3n3 linolenic acid	0.501 ± 0.15	0.781 ± 0.37	<0.05	C24:1 nervonic acid	0.402 ± 0.25	0.076 ± 0.1	<0.05
C18:4 stearidonic acid	0.058 ± 0.03	0.055 ± 0.02	NS				

**Table 8 ijerph-18-06500-t008:** Correlations between the results of neuropsychological tests and demographic factors and comorbidities.

Parameter	Neuropsychological Tests
BMI (kg/m^2^)	SC (r = −0.449) *
Age (years)	FDST (r = −0.38), 2 FDST (r = −0.401), BDST (r = −0.373), VFT-A (r = −0.0552), VFT-F&V (r = −0.37), VFT-K (r = −0.319), 2 VFT-F&V (r = −0.602), 2 VFT-K (r = −0.415) *
Diabetes/impaired fasting glucose (n)	VFT-F&V, LDFR **
Excessive drinking	VFT-A **
Smoking	2 List A **
Coronary heart disease	SDT A, 2 SDT A, VFT-F&V, VFT-K, VFT-P, 2 VFT-K, 2 VFT-P, 2 List A, 2 SDFR, 2 SDCR, 2 LDFR, 2 SC **
Hypertension	TMT A, 2 TMT A, 2 FDST, VFT-A, VFT-K **
Triglycerides (mg/dl)	BDST (r = −0.413) *
HDL cholesterol (mg/dl)	BDST (r = −0.535) *

* *p* < 0.05 (chi^2^ Pearson’s) ** *p* < 0.05 (Mann-Whitney U test), NS—statistically non-significant.

## Data Availability

The data presented in this study are available on request from the corresponding author. The data are not publicly available due to privacy restrictions.

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
