# Peer review of "Free Fatty Acids Are Associated with the Cognitive Functions in Stroke Survivors"

_ijerph, 2021, doi:10.3390/ijerph18126500_

Round 1

Reviewer 1 Report

Dear Authors,
I read your article with great interest. The chosen topic - the possible links between post-stroke cognitive impairment and free fatty acids (FFA) - is innovative and important. Because there is very little data on the potential impact of FFA on cognitive functions in stroke survivors, this topic is certainly worth further detailed studies, and your work deserves respect and attention. 
However, reading your manuscript, I have a number of questions and doubts, the answers to which I am convinced could significantly improve the importance and credibility of your work:

  1. The manuscript does not include any clinical characteristics of stroke (localization and size of infarction, neurological deficits according to NIHSS) that are important for post-stroke cognitive function.
  2. The introduction is probably too long, and some parts of it may be moved into discussion. Especially since some sentences and statements are repeated in the introduction and discussion (for example, lines 62-64 in Introduction and  lines 281-283 in Discussion).
  3. The inclusion/exclusion criteria of subjects - a) have you included patients with metabolic (e.g., hyponatremia, hypoglycemia-hyperglycemia) and hormonal (e.g., hypothyroidism) disorders that may affect cognitive function, and established cognitive impairment prior to a index stroke? 
  4. It is understood that fewer patients were examined during the follow-up visit than on day 7 after stroke. However, the difference (72 and 41 patients, respectively) is quite large. For what reasons did more than 1/3 of the patients (death, severe condition, refusal, lost-to-follow up?) not come to the follow-up visit and were there any attempts to contact them? 
  5. Figure 1 in the Results section is a bit confusing. It should at least be clarified which arrows in it imply a positive or negative correlation between the FFA and the CVLT. Do all metabolites in a particular branch have the same directional correlations with CVLT? 
  6. Only cross-sectional correlations between various FFA and cognitive tests at 7 days and 6 months are shown in the results. However, the change in patients' cognitive function between day 7 and month 6 after stroke is not presented ("Subgroup I included patients with MOCA test result < 26 (group I, 155 n=35), while subgroup II consisted of patients with MOCA score ≥ 26 points (group II, 156 n=37)" - lines 155-156. What MoCA estimates did the same patients have during the FU visit?)  Therefore, it is not possible to assess how FFA (and other factors not mentioned at all) may have affected cognitive functions. 
  7. You claim that "The results were given as sten values". As far as I'm familiar with sten values (or sten scores), STEN scores (or "Standard Tens") divide a scale into ten units (from 1 to 10) and represent ranges of values (https://psychology.wikia.org/wiki/Sten_scores; retrieved 2021-04-25). Is it really rational to calculate FFA correlations with STEN scores rather than with raw absolute numbers? 

Thank you.

Reviewer 2 Report

The subject of the manuscript does not fit in the scope of the IJERPH. The Authors might consider publishing the article in a journal in the field of biochemistry, metabolism or metabolomics after introducing necessary corrections. 

The main suggestions to improve the quality of the manuscript include:

  1. There should be a clear statement that the follow-up (N=41) group was significantly smaller than the initial group of patients (N=72). This information is missing in the abstract and in the results and may be found in 2.1 (p.3).
  2. The low statistical power as a result of the above should be discussed. Sten score should be explained and the use of such statistical approach should be justified.
  3. The introduction contains a lot of very detailed information and is not comprehensible to a reader without specialized training in biochemistry or lipidomics.
  4. There is an inconsistent use of terms. The authors state that the first evaluation was performed on day 7 after acute stroke onset, but later inform that the study was performed "on admission" or "at the stroke onset" (p.2).
  5.  The English language is appropriate, however some medical terms are not proper, e.g. "disturbances of consciousness" (p.3). 
  6. Material and Methods: 2.1 Subjects contains characteristics of patients with a list of medical problems - there is no further discussion of correlation of these factors with FFA profile. 2.2 Neuropsychological assessment lists a number of tests. It would be much more comprehensible if the Authors present the methodology in a table.
  7. Results are presented in tables with multiple numbers in columns and rows and it is hard to comprehend the significance of these results, e.g. Table 1 contains over 200 numbers and over 800 digits. Fig. 1 is unclear. FFA could be grouped to provide simpler presentation of the results.
  8. Discussion: the first part of the discussion repeats the results (v. 255-271). Discussion is somewhat incoherent and does not have a clear structure to guide the reader through the interpretation of the results.
  9. The limitation of the choice of day 7 of stroke for the FFA analysis should be discussed (e.g. in the light of metabolic and inflammatory changes associated with acute stroke).
  10. Analysis of correlations of the results with the nutritional status and blood cholesterol profile should be performed.

Reviewer 3 Report

Manuscript: Free fatty acids predict the cognitive functions in stroke survivors.

This is a very well written manuscript; overall information is well structured. I consider this manuscript presents valuable information about a novel topic. Authors present results from cognitive assessments and FFA measurements after stroke. I believe this manuscript is suitable for publication if minor improvements are made.

Specific comments follow:

  1. Title: Please consider substituting the word “predict”, the statement is too strong, given the fact that these findings are novel and more studies will be needed to confirm them.
  2. Line 119-122: Could you please specify if the medication list presented in this paragraph corresponds to the medications patients were previously taking and reported at baseline/ study enrollment or were they started during hospital stay? In the same paragraph, please consider using “hypotensive agents” instead of hypotensives.
  3. Were any of these patients taking Omega-3 supplements?
  4. Did you observe any trends on results based on patient’s demographics? Gender/age.
  5. Line 290: Please remove “the” before cognitive disorders.

Author Response

This manuscript is a resubmission of an earlier submission. The following is a list of the peer review reports and author responses from that submission.